# Analysis, Evaluation and Reusability of Virtual Laboratory Software Based on Conceptual Modeling and Conformance Checking †

Athanasios Sypsas *  and Dimitris Kalles 

School of Science and Technology, Hellenic Open University, 26335 Patras, Greece; kalles@eap.gr
* Correspondence: sypsas.athanasios@ac.eap.gr; Tel.: +30-69-4612-0759
† This paper is an extended version of our paper published in Modeling and Applied Simulation Conference 2022, Rome, Italy, 19–21 September 2022. https://doi.org/10.46354/i3m.2022.mas.002 (accessed on 20 March 2023).

**Abstract:** Virtual laboratories have been increasingly used in tertiary education for natural and applied sciences, especially due to the COVID pandemic, generating a substantial investment in corresponding software applications, including simulation experiments and procedures. However, it is expensive and time-consuming to analyze, understand, model and implement the virtual experiments, especially when it is necessary to create new ones from scratch, but also when they must be redesigned and addressed to an audience in a different educational setting. We use UML Activity Diagrams and Petri nets to model experimental procedures and then apply conformance checking to detect possible nonconformities between expected model behavior and actual model execution. As a result, we provide an estimation of the conceptual proximity between experiments performed in different educational settings using the same virtual laboratory software, assisting educators and developers in making informed decisions about software reuse and redesign by providing a systematic and formal way of evaluating software applicability. A virtual microscoping experiment was used as a case study for validation purposes. The results revealed that the specific virtual lab software can be ported, without modifications, from tertiary to secondary education, to achieve learning outcomes relevant to that education level, even though it was originally designed for a distance education university. The proposed framework has potential applications beyond virtual laboratories, as a general approach to process modeling and conformance checking to evaluate the similarity between the specification of experimental procedures and actual execution logs can be applied to various domains.

**Keywords:** conceptual modeling; virtual laboratory software; conformance checking

**MSC:** 68N30

## 1. Introduction

Education in natural and applied sciences is essentially based on laboratory practice, which provides the necessary skills for reinforcing concepts and hands-on learning [1,2]. Inquiry-based learning activities can provide valuable learning opportunities for students to improve their scientific literacy [3]. For a long time, natural and applied science experimentation and inquiry-based learning activities have been implemented in hands-on laboratories by asking learners to perform real experiments [4,5]. ICT advances have greatly contributed to the popularity of inquiry-based learning [6]. Simulation-based learning environments have great potential to improve students' knowledge of scientific subjects [7–9]. Virtual laboratories constitute a special category of simulations, offering the user the opportunity to conduct the same scientific inquiry provided by hands-on experiments but with the advantages of a virtual environment [10], as they are based on physical laboratory models and the experimental processes taking place therein [11,12]. During the

COVID-19 pandemic, practical sessions and experiments in schools and universities were suspended. However, they are necessary for students' skill development and experience in laboratory-based disciplines [13]. Thus, online education approaches gained popularity and quite a few educational institutions successfully adopted them [14,15]. Virtual labs have numerous advantages over traditional laboratory settings, including unlimited time, immediate feedback, the ability to repeat experiments, and safety for learners. In virtual labs, learners can conduct experiments at their own pace and without the constraints of limited lab time or access to expensive or rare equipment [16]. Subsequently, a substantial investment in corresponding virtual lab software applications has been made. In the following section, the evolution of virtual laboratories software is presented.

### 1.1. Virtual Laboratories Software

Virtual laboratories software development tools are evolving over time and new methods for modeling physical laboratories and procedures are being adopted. The initial approaches used computer simulation software stored and executed on local workstations. Virtual lab exercises were developed by enterprises or educators and designed for specific experiments [17,18]. Additionally, they were locked, not giving educators the ability to alter the parameters of the experimental process. Following this, virtual labs were developed as web-based applications, distributed VLs, which could be executed as games [19]. Due to the initially slow Internet speeds, the user interface was 2D and not as detailed. Some of the tools included Macromedia flash [20], Macromedia Authorware [21], Microsoft VB5 [22], LabVIEW, a powerful graphic environment for the development of virtual instruments and OOP languages such as C++, Java or C# [23–25]. The recent progress in access to the Internet, as well as in 3D graphics and software packages, enabled the exact representation of physical labs in a computer-simulated environment [26]. The virtual laboratory software named Labster (www.labster.com, accessed on 28 February 2023) is widely used [27,28], while other researchers use Unity 3D $^{TM}$ and Unreal Engine 4 $^{TM}$ to build an appropriate VR environment [29–31]. The kinetics$^{TM}$ and MoDS$^{TM}$ developed by CMCL Innovations (https://cmclinnovations.com/, accessed on 28 February 2023) were used for virtual chemical engineering labs [32]. Some of the virtual labs can be used in tablets and mobile phones, facilitating anytime, anyplace access [27–29]. Moreover, new applications offered educators the ability to use already-tested experiments or create new ones, adapted to various educational conditions. However, creating virtual experiments from scratch or redesigning them to address an audience in a different educational setting is expensive and time-consuming. Therefore, the reuse of implemented experiments within specific simulators could offer an alternative solution. To effectively model activity in integrated systems, researchers can leverage approaches from prior work. Since 1969, standards for describing models and simulations have been advocated [33]. However, to establish these standards, model descriptions must be stored and exchanged in a way that enables efficient reuse [34]. Standardization is a crucial component in enabling the exchange and interpretation of scientific research outcomes, especially in computational modeling [35]. Thus, a formal description of simulation experiments is needed to allow for similar experiments to be addressed to a different audience of learners [32]. Additionally, establishing complex virtual laboratory software systems that are custom-designed for education and will need to be maintained can be a costly undertaking and thus requires a disciplined approach to development based on software engineering models and methods.

### 1.2. Software Engineering Models and Methods

The goal of software engineering (SE) is to provide models and processes that lead to the production of well-designed, well-documented, and maintainable software products [36]. Software engineering models are the various processes or methodologies that are selected and used for the development of software products. They provide a structured and organized way of designing, implementing, and maintaining software products. Each of these models has its own set of characteristics, advantages, and limitations, and is used

in different situations depending on the project's requirements, size, complexity, and other factors. In SE, understanding and modeling the environment is an important part of system development [37]. Conceptual modeling is used in almost all of these approaches, i.e., in the waterfall model, it is used in the analysis stage [38], while in agile software development it can be used for knowledge management [39]. Moreover, software analysis is at the heart of quality software development. Part of this analysis aims to clarify what has already been implemented and how to adapt software that already exist under the constraints of the new domain. SE aims to reuse as many previously developed software artifacts as possible, since software reuse can provide a better means of SE [40,41]. Furthermore, software reusability can be seen as a new approach for SE, leading to less development time, high-quality results, a reduction in software maintainability, less effort for developers and cost-effective solutions [42]. To calculate the cost, an estimation must be made of the extent of the relationship between what we want to implement and what already exists. By providing the necessary information to software designers during the design and implementation phases of SE, they can decide whether an application will be built from scratch or if a previously implemented software product will be reused. Moreover, model-driven software engineering (MDSE) uses modeling for the development of software artifacts [43]. MDSE practices are proved to increase effectiveness and efficiency in software development [44]. Furthermore, model-checking validates the behavior of a given system when modeled by a finite state machine or other modeling notation [45]. Therefore, detecting differences between different versions of models is important because it contributes to the further analysis and comprehension of a system, which can help it to evolve and identify possible alternatives. By comparing and analyzing different versions of a model, researchers can identify changes and improvements that have been made, as well as areas in which further work may be necessary [46]. Software process models used in SE often represent a sequence of activities, objects, transformations, and events that embody software evolution strategies [47]. Nevertheless, software process models may differ from the executed processes when software is run. Therefore, collecting data through logging from successive runs based on certain process models allow for us to think about how to analyze and possibly improve the models.

*1.3. Conformance Checking*

Process mining (PM) techniques are used to study when and how a specific process deviates from the process model. Based on the extracted knowledge from event data (i.e., event logs) produced by information systems, PM is mainly used to analyze, discover, and enhance processes [48–50]. PM has been successfully applied in various areas, including software engineering [51,52]. Therefore, process mining techniques are applied to find development processes using publicly available, open-source, software development repositories [53]; in another study, a process mining framework was used to discover software development processes from the event logs generated by software configuration management (SCM) systems [54]. Software houses with complex software development processes are appropriate for the application of PM techniques. In such organizations, there may be obstacles to the working processes, making them difficult to complete [55]. By using PM, the conformance of the resulted process model is verified and useful process-oriented information is retrieved. However, many studies on PM focus on describing the PM techniques, demonstrating their efficiency in discovering software process models, leaving aside the software development process evaluation [52]. As a conclusion, PM can be used during the actual software development process to generate new models or confront the existing models with reality. Conformance checking contributes to the decision as to whether the execution of the process conforms to the corresponding process model [56]. In study [57], two metrics concerning conformance checking were introduced: fitness and appropriateness. Fitness is used to measure the degree to which the process model can replay the traces from the stored event log file [48].

*1.4. Contribution*

Conformance checking is a fundamental branch of process mining (PM) that involves the comparison of a process model with real-life process executions to identify potential deviations, bottlenecks, and inefficiencies in the process [58]. The goal of this branch is to explain and quantify deviations in a non-ambiguous manner, drawing conclusions concerning the process model and real execution data correspondence [59]. Hence, as the calculation of "deviation" necessarily involves some amount of "similarity" calculations, conformance checking could serve as a soft computing technique for virtual lab software analysis, evaluation and possible reusability, since it is based on process models describing the real lab experiments. To the best of our knowledge, there is currently no study that suggests a combination of conceptual modeling and conformance checking to deal with the analysis and reusability of virtual laboratory software. This paper aims to propose a framework that can be used under a software engineering process, contributing to problem analysis, and the understanding and assessment of the possible applicability of a solution. It incorporates soft computing techniques at various stages in order to decide whether to use a new simulator or reuse one that already exists. To achieve this goal, we can utilize standard tools from the conceptual modeling repertoire, such as unified modeling language (UML) activity diagrams (ADs) and Petri nets (PNs), to formally describe the experimental procedure. UML ADs and PNs are powerful modeling languages that can represent both the static and dynamic aspects of an experimental procedure, as well as composite flows. By using these tools, researchers can create a comprehensive and accurate representation of the experimental procedure, which can be shared and analyzed by others. After obtaining simulation traces from the corresponding Petri nets and logged data from virtual experiments executed in a specific virtual laboratory, Onlabs (https://sites.google.com/site/onlabseap, accessed on 15 February 2023), the conformance checking metric of fitness can be calculated. This metric quantifies the degree to which the executed experiments conform to the expected behavior described by the Petri nets. The proposed method uses the simulated processes for conformance checking and is independent of any process model notation. Our approach aims to analyze and evaluate the virtual laboratory software, offering an estimation regarding the conceptual proximity of the same experiments when performed in diverse educational settings. Consequently, the suggested framework empowers educators to determine whether a specific software application can be reused with minimal modifications or if it needs to be redesigned. A virtual microscoping experiment is used to validate our approach. The suggested framework concerns a wide and growing field of applications, as it may serve as an essential part of software analysis and reusability.

The remainder of the paper is structured as follows. In Section 2, we present the conceptual modeling of experiments in the virtual laboratory software. In Section 3, we describe our approach based on conformance checking and alignment techniques. Section 4 outlines the application of our approach and the experiment used for validation, followed by the results and a discussion of their interpretation in the context of previous studies and our hypotheses. Section 5 concludes the paper by summarizing our findings, acknowledging the limitations of our research, and providing future applications for our proposed approach.

## 2. Conceptual Modeling of Virtual Laboratory Experiments

Conceptual modeling is a technique used in software engineering to create a high-level abstract representation of a software system [60]. The goal of conceptual modeling is to capture the key features and functionality of a software system in a way that is understandable to both developers and stakeholders. This is typically achieved using visual models, such as diagrams, that can help to clarify composite relationships, flows and interactions within the system. Moreover, conceptual modeling is an important part of the software development process, as it can help to identify potential problems early on and ensure that the final system meets the needs set before implementation. The conceptual

modeling of experiments executed in a virtual laboratory involves the creation of a high-level representation of the experiment, its components, and their interactions in a virtual environment. The goal is to provide a clear and comprehensive understanding of the experiment's purpose, design, and expected outcomes.

### 2.1. Conceptual Modeling Tools

In the following sections, two of the most-used tools in conceptual modeling repertoire are presented. They are used for the analysis of the virtual laboratory software under study.

### 2.1.1. UML Activity Diagrams

Unified modeling language (UML) is a standardized modeling language that can be used across different programming languages and development processes. Specifically, UML activity diagrams (ADs) constitute a powerful tool because they can model the composite flows and sequence of actions, thus capturing the process flow and its results [61,62]. Additionally, Ads are adopted as a standard tool in the IT industry to model workflows and investigate system behavior [63]. Depending on their actual semantics, ADs are a mixture of dataflow diagrams, and Petri nets [62]. Additionally, they are mostly used to describe the flow of activities in a wide range of settings, including computer systems, business processes, use-case processes, experimental processes, and serious educational games [64,65].

Models for virtual laboratories continuously evolve to meet the educational needs of a variety of educational settings. Thus, a formal description of simulation experiments is needed [66]. Description languages, diagrams, and visual modeling environments and tools are commonly used in experimental processes to describe the procedures and steps involved in the process, as well as the data objects that are needed [67]. Subsequently, UML ADs describing virtual experiments can be mapped, using the appropriate tool, to any concrete implementation language, such as C or Java, so that they can be embedded in any virtual laboratory environment. However, in order to be reused in different simulation environments, rules for simulation experiments should apply, including a precise description of the simulation steps and other procedures [68].

An AD can be described as [69]

$$AD = \langle A, V^{inp}, V^{loc}, AN, PN, T \rangle$$

where

- $A$ is a set of action names;
- $V^{inp}$ is a set of input variables over finite domains;
- $V^{loc}$ is a set of local variables over finite domains;
- $AN$ is a set of action nodes, $an_1$ with acname (an) = ac $\in$ A;
- $PN$ is a set of pseudo-nodes, such as initial nodes $PN^{init}$, final nodes $PN^{fin}$, decision nodes $PN^{dec}$.

$T$ is a set of transitions of the form $t = \langle n_{src}, n_{trg}, \text{guard} \rangle$, where $n_{src}, n_{trg} \in (AN \cup PN)$ and guard is a Boolean expression.

While activity diagrams provide a useful graphical representation of the behavior of a system, they do not support formal analysis in the same way that Petri nets or automata do. AD-based specifications are considered semi-formal because they lack the tools necessary for rigorous analysis and verification before implementation. To address this limitation, ADs can be transformed into a formal specification such as Petri nets or automata, which can then be analyzed using verification tools [70].

### 2.1.2. Petri Nets

Petri nets (PN) are a well-established mathematical modeling language for the description of various systems [71] and for modeling processes in several domains [57]. In addition, they are a graphical and mathematical modeling tool applicable to many systems.

They are frequently used to describe and study information processing systems that are characterized as asynchronous, concurrent, parallel, distributed, nondeterministic, and/or stochastic [72].

A PN without any initial marking is represented as a three-tuple $N = \{P,T,F\}$, where

$$P = \{p_1, p_2, \dots , p_n\} \text{ is a finite set of places;}$$

$$T = \{t_1, t_2, \dots , t_n\} \text{ is a finite set of transitions;}$$

$$F \subseteq (P \times T) \cup (T \times P) \text{ is a set of arcs.}$$

The set $(P \times T)$ represents the directed arcs from T to P, while set $(T \times P)$ includes the directed arcs from P to T, respectively. The behavior of many systems can be described in terms of states and their changes. Although Petri nets are simple and graphical, they allow for the modeling of concurrency, choices, and iteration. Thus, the syntax of PN and AD has to be reviewed in order to define a formal mapping between them. Prior work has provided mapping rules from ADs to PN [63,73], based on an equivalence of the modeling elements in PN and ADs. As a result, we mapped the ADs abstractly describing the experiments executed in a virtual lab into the equivalent PN.

### 2.2. Virtual Laboratory

The virtual laboratory environment utilized in this study is the Onlabs software (https://sites.google.com/site/onlabseap/, accessed on 15 February 2023), developed by an interdisciplinary team at the Hellenic Open University (HOU), a distance education institution specializing in distance learning education. Figure 1 depicts the virtual laboratory environment.

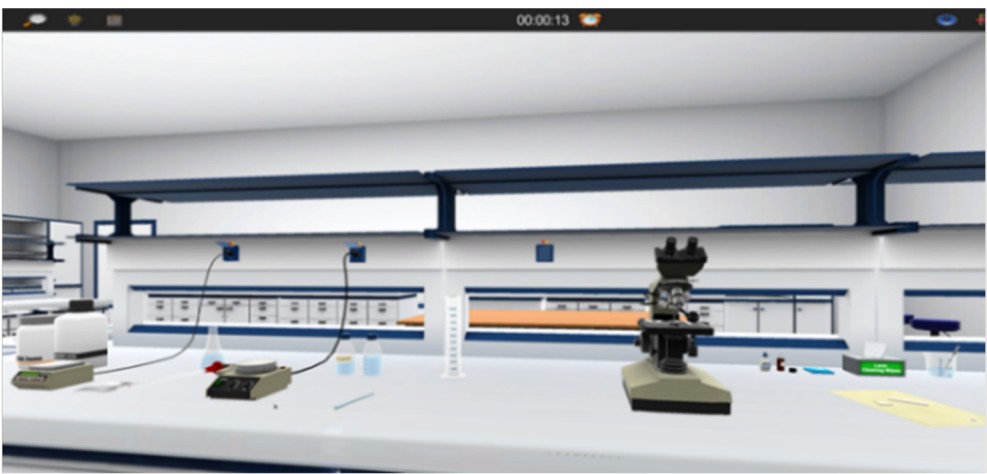

**Figure 1.** The Onlabs virtual environment.

The Onlabs software provides a virtual laboratory environment where users can navigate and manipulate instruments to conduct experiments. It is designed to be an applied research laboratory simulator using powerful game engine, Unity, to enable multi-platform support. The latest version of Onlabs includes experiments for microscoping a test specimen and preparing an aqua solution, with the latter featuring quantitative aspects of the simulation.

### 2.3. Modeling Experiments within a Virtual Laboratory

UML ADs are used to model the experimental procedure of microscoping for HOU and secondary education. In Figure 2 below, an AD is presentes depicting the initial steps of the experimental procedure of microscoping at the HOU level.

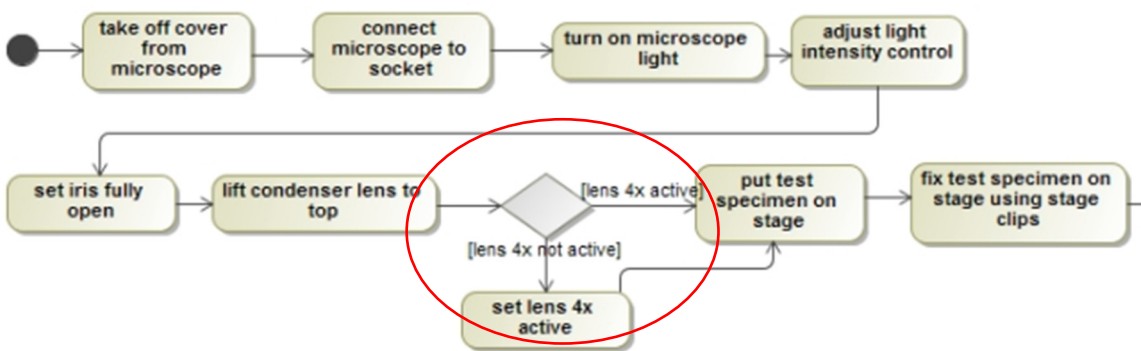

**Figure 2.** Part of the AD describing the experimental procedure in HOU.

In Figure 3, the corresponding steps of the experimental procedure at the secondary education level are depicted.

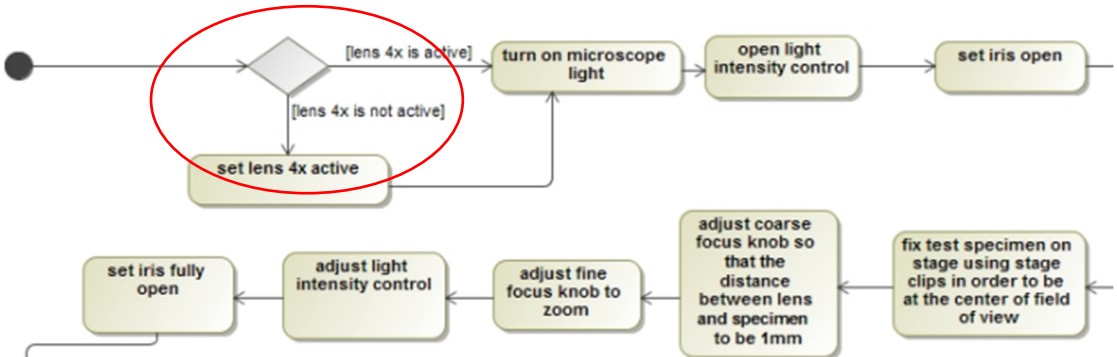

**Figure 3.** Part of the AD describing the experimental procedure in secondary education.

The experimental procedures are depicted in UML ADs, which are a widely used and easily understood modeling language. The choice of UML activity diagrams (ADs) to model the experiments for the virtual lab was motivated by the need to involve scientists from different domains, including biology, chemistry, and computer science. UML is a standardized notation that is widely recognized and understood by researchers from different fields, making it an ideal choice for interdisciplinary collaboration. The interdisciplinary team working on the virtual lab design needs to understand the underlined model in order to analyze and evaluate it. By studying and comparing these models, a further analysis was made regarding the application of a virtual lab to a different audience.

To increase the level of formality in the conceptual modeling of experiments in Onlabs, Petri nets were used to model the same experiments. Therefore, in Figure 4, part of the corresponding PN of the microscoping experimental procedure in HOU is presented. The PN modeling of the experiment in secondary education is shown in Figure 5.

Therefore, PN tools such as Yasper and PIPE v. 4.3 can be used for PN verification and simulation, leading to useful results. These tools use an extended Petri net as a modeling technique. Additionally, they offer easy editing, token gameplay and performance analysis with a randomized automatic simulation for basic place/transition nets. They also offer case-specific vs. inter-case token flow, decision nodes with parameterized probabilities of alternatives, and reset and inhibitor arcs [74,75]. Then, the model represented via PN can be compared to the model of the experiment used in another educational setting to decide whether it can be reused.

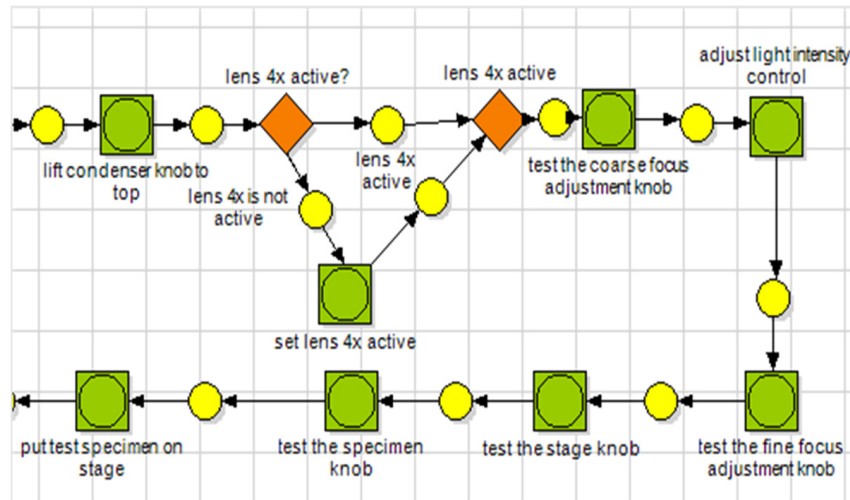

**Figure 4.** Part of the PN of microscoping experiment in HOU.

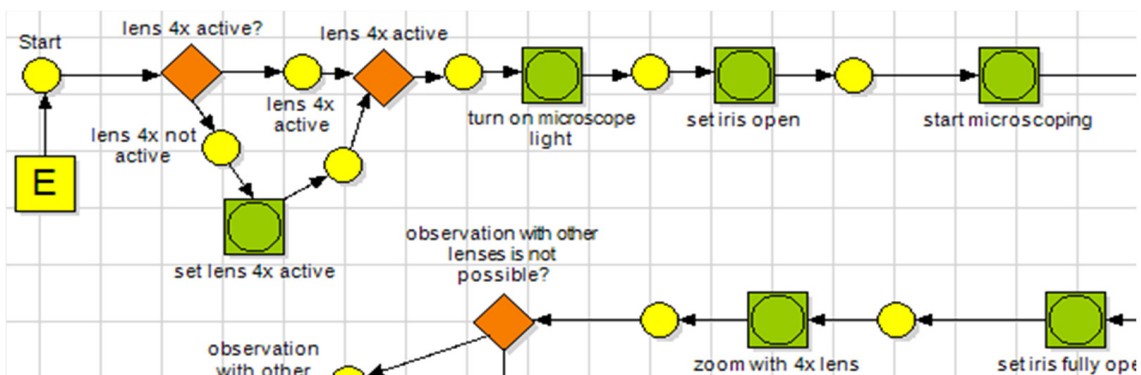

**Figure 5.** Part of the PN modeling the microscoping experiment in secondary education.

## 3. Conformance Checking Framework for Evaluation and Reusability of Virtual Laboratory

The information and communication applications used in various institutions can log the educational processes that take place therein. Taking advantage of these logged data, a research area, Educational Process Mining (EPM), aims to monitor, discover, and improve educational processes by extracting valuable information from stored event logs [76]. In the context of virtual laboratories, process models can be used to describe the sequence of necessary actions that lead to a specific learning outcome. The further analysis of these processes contributes to the deeper comprehension of experiments and learning flows, since the logged learners' actions are associated with the process models describing the standardized experimental procedures [48]. These process models can be compared to the logged events to identify deviations between the model and the reality. This is achieved through a technique called conformance checking, which computes an alignment value to compare process models [56]. By analyzing these deviations, researchers can gain a deeper understanding of the learning flows and improve the virtual laboratory experience. The standard conformance checking technique is alignments computation [77].

### 3.1. Conformance Checking Preliminaries

Using the conformance checking technique, the stored event logs and the model are compared by calculating the alignment between the observed and modeled behaviors. As a first step, any logged events not associated with any task in the model are removed. Then, the execution traces are stored in an event log, and the potential variations between traces

and the process model are evaluated [78]. The main type of conformance checking metric is Fitness [57].

Fitness

The fitness metric is important to estimate the severity of potential deviations and to compare different model–log combinations. To compute the alignment, the moves (tasks) in the process model need to be associated with the moves (events) in the trace contained in the log file. To establish a correspondence between the tasks in the model describing the experiment and the logged events, a label denoting the associated log event type needs to be assigned to each task in the model [79]. For example, a Petri net with labels $l$ of process model tasks/actions $t$ named

$$l(t_1) = k_1, \ l(t_2) = k_2, \ l(t_3) = k_3, \ l(t_4) = k_4, \ l(t_5) = k_5, \ where \ k_1, k_2, k_3, k_4, k_5$$

represent the events in the stored log files and a stored trace

$$\sigma = k_1 \ k_2 \ k_3 \ k_2 \ k_4 \ k_5.$$

Possible alignments are presented below.

$$\gamma_1 = \left| \begin{array}{c} k_1|k_2|k_3|k_5 \\ t_1|t_3|\perp|t_5 \end{array} \right|, \gamma_2 = \left| \begin{array}{c} k_1|k_1|k_2|k_5 \\ \perp|t_1|t_2|t_5 \end{array} \right|, \gamma_3 = \left| \begin{array}{c} k_2|k_2|k_3|k_4 \\ \perp|t_1|t_3|t_4 \end{array} \right|$$

The top row refers to the trace stored in the event log, while the bottom row refers to the process model tasks. Notation $\perp$ is used to describe the nonappearance of a model task even when an event is stored. Therefore, the first move in $\gamma_3$ is $(k_2, \perp)$, declares that when a trace action $k_2$ is carried out, no task is executed in the process model. In a labeled Petri net, the moves can be categorized as synchronous, log-only, model-only, or illegal, based on the matching of tasks and events during the conformance checking process.

A synchronous move occurs when both the model and log move simultaneously, meaning that the observed behavior matches the expected behavior according to the model. A log-only move occurs when only the log moves, meaning that an event is observed that is not captured by the model. This could indicate an error in the model, or a real-world behavior that was not anticipated during the modeling phase. A model-only move occurs when only the model moves, meaning that an action is taken in the model that is not observed in the log data. This could indicate an error in the log data, or a difference between the simulated and actual behavior of the system. An illegal move occurs when there is a mismatch between the model and the log data, indicating a violation of the expected behavior. This could be caused by errors in the model, errors in the log data, or discrepancies between the simulated and actual behavior of the system [79]. Successively, the fitness as part of alignment can be defined as:

Fitness = (events in trace mimicked by the model)/(total moves in the observed trace)

The closer the fitness value is to 1, the more similar the model is to the event log file.

*3.2. Proposed Implementation Framework*

The proposed implementation framework is based on a comparison of the traces of virtual lab log file and the PN of the same experiment. The virtual lab experiment execution and the PN are from different educational settings. Thus, the results of the comparison can be used to determine whether the same virtual lab software can be reused in a variety of educational settings.

As the first step in our approach, we model experiments for both university and secondary education levels using UML activity diagrams (Ads) and the corresponding Petri nets. Afterwards, using Yasper tool, we validate, simulate and collect the simulation traces of the PN describing the experiment in one educational setting. In order to calculate

the fitness metric, the experiment is performed in the Onlabs virtual laboratory in a different educational setting, and the ordered tasks are recorded in a log file. These traces are then used to compute the fitness between the model and the log. First, the tasks in the process model are associated with the logged events by using labels to denote the associated log event type for each task in the model. The mapping results are stored in an association database containing the associated model events and logged traces required for alignment computation. The mapping is then used for fitness calculation. The fitness results can help determine if the same virtual laboratory can be reused for the experiment in a different educational setting without further study.

Figure 6 illustrates the proposed implementation framework.

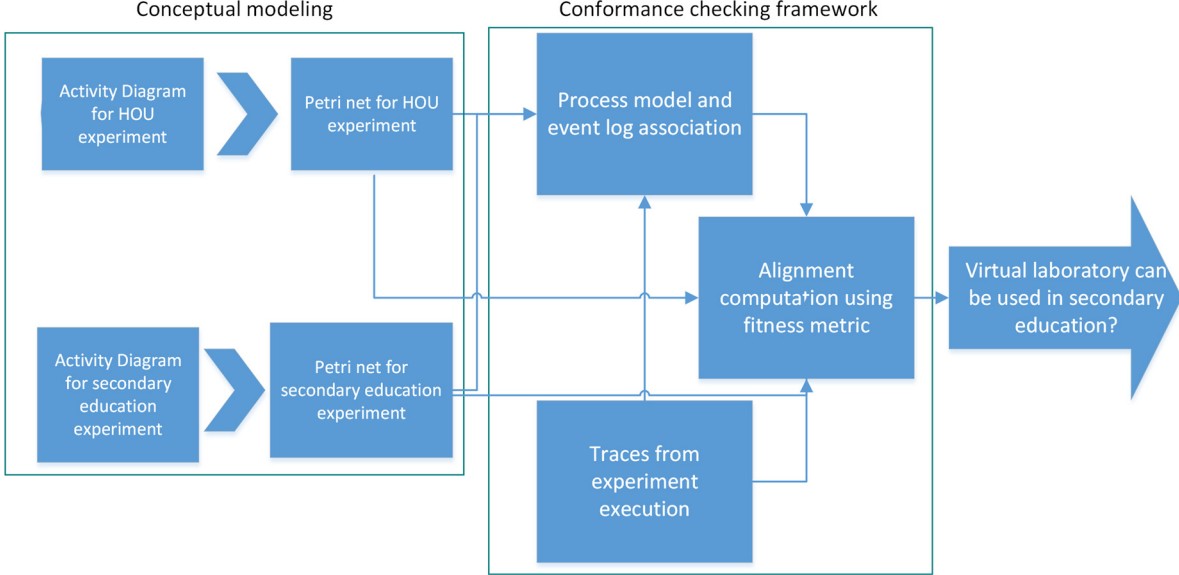

**Figure 6.** Proposed implementation framework.

## 4. A Prototype Implementation of a Fitness Metric for Virtual Lab Experiments to Assist Education Analysts

The proposed implementation was tested by an experiment in secondary school in Preveza, Greece. The participants were 25 first-year high school students in a biology class. The students had to execute the microscoping experiment in the virtual laboratory (Onlabs) without prior experience. They formed teams of two or three students per computer and were provided with written instructions describing the experimental steps of the HOU microscoping experiment. The educational activity lasted one teaching hour (40 min). At the start of the activity, the microscope was displayed prominently on the computer screen due to time constraints. Initially, the students were offered some explanations about the purpose of the activity, to allow them to become familiar with the microscope's function. The students handled the microscope's functions easily and showed interest and enthusiasm during the activity despite the problems with some computers. They even managed to solve the problems on their own, such as by using the space key to "release" the screen or stopping the virtual laboratory rotations that occurred against their will. Cooperation and mutual assistance were developed among the groups. Some groups successfully faced and solved a problem, while other students faced a different one, and they exchanged their views. Since the experimental process was not guided by the software, students followed the experimental steps of their own free will. Help was only requested from the instructor in rare cases. The instructor monitored the activity and intervened only when necessary. At the end of the activity, the instructor asked the students for feedback and summarised the lesson's objectives.

Then, log files were selected from each computer for further analysis and conformance checking. The proposed approach was applied to the processed log files and the fitness values are shown in Table 1.

**Table 1.** Results of fitness calculation.

| Log File | Fitness Value |
|---|---|
| Log1 | 0.59 |
| Log2 | 0.57 |
| Log3 | 0.61 |
| Log4 | 0.63 |
| Log5 | 0.56 |
| Log6 | 0.55 |
| Log7 | 0.62 |
| Log8 | 0.65 |
| Log9 | 0.58 |
| Log10 | 0.67 |
| Log11 | 0.58 |

The above table shows the fitness calculation values for 11 different log files, with values ranging from 0.55 to 0.67. It is important to note that fitness calculation values are not the only measure of the effectiveness of a simulator in a specific educational setting. The fact that all students were able to complete the experiment successfully is a strong indication that the simulator can be effectively reused in secondary education, without additional preparation. However, it is also important to consider other factors, such as student engagement, learning outcomes, and satisfaction with the learning experience, to determine the overall effectiveness of a simulator in a particular educational setting. Most of the tasks in the process model for the HOU microscoping experiment were also observed in the logged events from the virtual laboratory application in secondary education. However, the order of some model tasks differed, indicating the need for some alignment. For instance, the decision node on whether the microscope lens 4x is active was in a different order between the model event for the secondary education experiment and the logged trace from the university experiment. Figures 7 and 8 depict the differences between UML AD for secondary education experiment and the corresponding steps included in the log file from the HOU experiment's execution.

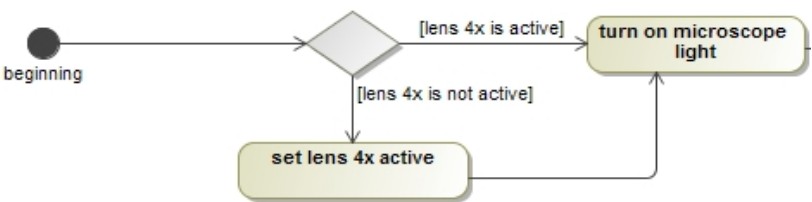

**Figure 7.** Part of the UML AD for secondary education experiment.

```
used mic_plug with socket1
set mic_AC_switch on
rotated mic_light_intensity_knob
rotated mic_aperture_knob
rotated mic_aperture_knob
rotated mic_condenser_knob
```

**Figure 8.** Part of the log file for HOU experiment.

Nonetheless, it can be concluded that the general learning outcome of manipulating microscope lenses, as set by the educational program, was achieved. Whether the learn-

ing outcome was achieved depends on the educator's opinion, considering the fitness calculation. Despite the fact that the virtual lab was originally designed for university students, in the experiment described in the previous section, lower secondary education students completed the microscoping experiment. Although students achieved rather small fitness, the above alignment calculation results revealed that the microscoping experiment can be used in secondary education by reusing the same virtual laboratory designed for university-level education. This is in line with the findings of Sypsas and Kalles (2022). In summary, our proposed approach allows learners from different educational settings to benefit from educational tools and resources commonly used in university distance education. If necessary, educators can modify the experimental steps to achieve the desired learning outcomes, and the process model can be adjusted accordingly.

Another study used event process logs from a custom software development process within the IT department of a large automotive company to extract information about the working process model, organizational network, and statistical information [55]. Based on the obtained results, an action plan was developed to improve the development process and the release result. This approach is similar to other studies that use event logs to extract information and improve processes and demonstrates the potential of using data-driven insights to optimize software development processes.

Thus, a central soft-computing technique, that of computing similarities, now appears as a process alignment computation to analyze and evaluate the suitability of virtual laboratory software. Additionally, it offers instructors an indication of the reusability of the software in diverse educational settings.

However, there are still limitations to our approach. Although users from lower secondary education were included in the validation check, the promising results must be viewed in the context of our approach being applied to a single experiment. Therefore, further research on other experiments in the same virtual lab, executed in various educational settings, is necessary to validate our approach.

## 5. Conclusions

In software engineering, it is crucial to understand and model the environment in which a system will be used. This includes factors such as user requirements, organizational processes, hardware and software infrastructure, and other external factors that may impact the system's performance and functionality. By modeling the environment, developers can better anticipate potential issues and design solutions that are better suited to the specific context in which the system will operate. Developing virtual experiments from scratch or redesigning them for different educational settings can be a time-consuming and costly process. Therefore, the ability to reuse existing virtual experiments across different educational settings with minimal modifications can be beneficial in terms of time and resources. In pursuit of this goal, we proposed a framework using conceptual modeling for the analysis and better comprehension of the virtual lab experiments. The proposed approach uses UML activity diagrams and Petri nets to model the experimental procedure, serving as appropriate tools for developers to work on the existing virtual lab software. Subsequently, we used the conformance checking technique to detect deviations between expected model behavior and actual model execution. Therefore, the actual stored log files of the investigated system were used to compute a similarity index by aligning the model of an experiment at a specific educational level with the logged file produced from a simulation of the same experiment at a different educational level. The ultimate goal is to provide educators with an estimation of the conceptual proximity between experiments performed in different educational settings, enabling them to determine whether a virtual laboratory software application can be reused with minimal modifications or requires a redesign. Finally, our approach was applied, and the results revealed a good performance in well-defined experiments, concluding that they can be executed in different educational settings by reusing the same virtual laboratory environment, without further investigation.

In order to tackle the limitations of our present study, we will use the proposed approach to validation in other experiments that are commonly executed in a variety of educational settings, all with possible changes (the actual experiment we are working on is the production of an aqueous solution), trying to reuse the virtual laboratory software.

The proposed framework has potential applications beyond virtual laboratories and can be used in recommendation systems and software analysis and reusability. This is because the use of process modeling and conformance checking to evaluate the similarity between experimental procedures and execution logs can be applied to various software applications beyond virtual laboratories. By providing a systematic and formal way of evaluating software applicability, the proposed framework can help educators and developers to make informed decisions about software reuse and redesign.

**Author Contributions:** Writing—original draft preparation, A.S. and D.K.; writing—review and editing, A.S. and D.K.; framework implementation, A.S; supervision, D.K. All authors have read and agreed to the published version of the manuscript.

**Funding:** Part of this research was funded by the European Union (European Social Fund—ESF) and Greek National Funds through the Operational Program "The Human Resources Development, Education and Lifelong Learning Program (Digi-Lab, 70381)".

**Institutional Review Board Statement:** Not applicable.

**Informed Consent Statement:** Informed consent was obtained from all subjects involved in the study.

**Data Availability Statement:** Not applicable.

**Acknowledgments:** The authors would like to thank the secondary education science instructor who supervised the virtual lab experiment.

**Conflicts of Interest:** The authors declare no conflict of interest.

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
