# Peer review of "Analysis, Evaluation and Reusability of Virtual Laboratory Software Based on Conceptual Modeling and Conformance Checkingâ€"

_mathematics, doi:10.3390/math11092153_

Round 1
Reviewer 1 Report
The work presented has a more pedagogical orientation than a mathematical or computational one. The topic is interesting, but the level of mathematical and computational input is very low. The work is very descriptive and, from my point of view, does not fall within the scope of the journal, nor of the special issue. However, I consider that it would fit better in educational journals, as is also reflected in the references used.
Author Response
The work is based on a virtual laboratory for biology education. The analysis, evaluation and possible reusability of the software includes educational aspects as it is designed for learners. However, the presented approach has potential applications beyond virtual laboratories and can be used in recommendation systems and software analysis and reusability. This is because the approach of using process modeling and conformance checking to evaluate the similarity between experimental procedures and execution logs can be applied to various software applications beyond virtual laboratories. By providing a systematic and formal way of evaluating software applicability, the proposed framework can help educators and developers in making informed decisions about software reuse and redesign. For example, a similar approach is used in software development process within an IT department of a large automotive company (see reference [55] in the revised paper).

Reviewer 2 Report
The authors are conducting very interesting research with very good potential.
I recommend a set of improvements to the authors:
- Clarifying the data collected and the results obtained through their processing, especially the processing of logs. In its current form, the results provided by the authors are as follows: “The proposed approach was applied on the processed log files and the fitness values were as follow: 0.59, 0.57, 0.61, 0.63, 0.56, 0.55, 0.62, 0.65, 0.58, 0.67, 0.58.” In its current form, the importance of the values is not presented explicitly enough.
- Clarification of the relationship between the results obtained and the conclusions generated.
- Quality improvement of figures. Some are difficult to read.
Author Response
The study results are presented in a table format, which includes the log files used for analysis and the corresponding fitness calculation values that were derived from these logs.
An extra paragraph is added just after the presentation of the results, to clarify the relationship between results and conclusions.
An extra paragraph is added just after the presentation of the results, to clarify the relationship between results and conclusions.

Reviewer 3 Report
The main contribution of this paper lies in introducing a framework that enables to analyse and evaluate the virtual laboratory software and determine whether a specific software application can be reused with minimum modifications or if it needs to be redesigned.
I consider the topic original. It addresses the research gap as there is currently no study that suggests a combination of conceptual modelling and conformance checking to deal with the analysis and reusability of virtual laboratory software. The proposed framework has potential applications beyond virtual laboratories and can be used in recommendation systems and software analysis and reusability.
Paper is well written and understandable.
References are appropriate and sufficient.
Figure 6 is not readable.
There is a missing space in the title
2.3 Modeling experiments withinvirtual laboratory
Author Response
Figure 6 (proposed implementation) was redesigned in order to be readable and understandable.
The extra space in section 2.3 title is added.

Round 2
Reviewer 1 Report
The article submitted has improved on the initial submission in a few respects.
In my opinion, current soft computing has a set of very specific techniques, such as fuzzy systems, artificial neural or evolutionary computing. Therefore, I maintain my previous criterion, in the sense of considering that it does not fall within the scope of the journal, even with the nuance of the special issue.
I also maintain my criterion in the sense that it does not incorporate methodologies, techniques and tools with high mathematical intensity.
However, I broaden the scope of journals in which its publication may be of interest, specifically, to journals specialized in soft computing.